



# Virtual decoupling to break the Simplification Versus Resolution Trade Off in NMR of Complex Metabolic Mixtures

Cyril Charlier[1*], Neil Cox[1], Sophie Martine Prud'homme[2,3], Alain Geffard[2], Jean-Marc Nuzillard[4],
Burkhard Luy[5], Guy Lippens[1*]

[1] Toulouse Biotechnology Institute (TBI), Université de Toulouse, CNRS, INRA, INSA, Toulouse, France
[2] Université de Reims Champagne-Ardenne (URCA), UMR-I 02 SEBIO (Stress Environnementaux et Biosurveillance des milieux aquatiques), Moulin de la Housse, Reims, France.
[3] Université de Lorraine, CNRS, LIEC, F-57000, Metz, France (present address)
[4] Université de Reims Champagne Ardenne, CNRS, ICMR UMR 7312, 51097 Reims, France
[5] Institute for Biological Interfaces 4 – Magnetic Resonance, Karlsruhe Institute of Technology (KIT), Herrmann-von-Helmholtz-Platz 1, 76344 Eggenstein-Leopoldshafen, Germany.

*Correspondence to*: Cyril Charlier (charlier@insa-toulouse.fr) and Guy Lippens (glippens@insa-toulouse.fr)

**Abstract.** The HSQC experiment developed by Bodenhausen and Ruben (Bodenhausen and Ruben, 1980) in the early days of modern NMR is without a doubt one of the most widely used experiments, with applications in almost every aspect of NMR including metabolomics. Acquiring this experiment however always implies a trade-off: simplification versus resolution. Here, we present a method that artificially lifts this barrier, and demonstrate its application towards metabolite identification in a complex mixture. Based on the measurement of CLean In-Phase and CLean Anti-Phase (CLIP/CLAP) HSQC spectra (Enthart et al., 2008), we construct a virtually decoupled HSQC (vd-HSQC) spectrum that maintains the highest possible resolution in the proton dimension. Combining this vd-HSQC spectrum with a J-resolved spectrum (Pell and Keeler, 2007) provides useful information for the one-dimensional proton spectrum assignment and for the identification of metabolites in *Dreissena polymorpha* (Prud'homme et al., 2020).

## 1. Introduction

The recognition that a given nucleus was characterized by a specific chemical shift value depending on its exact environment in a molecule (Proctor and Yu, 1950; Dickinson, 1950) ushered Nuclear Magnetic Resonance (NMR) from its initial discovery in a nuclear physics department (Purcell et al., 1946; Bloch et al., 1946) into the chemistry sphere. Magnetic shielding of nuclei was identified at the origin of the phenomenon (Ramsey, 1950) and representative chemical shift values of the different protons in organic molecules were rapidly established (Arnold et al., 1951; Bernstein and Schneider, 1956). The further recognition of homonuclear scalar coupling (the « J-coupling ») patterns (Gutowsky and McCall, 1951) increased the information content, and notions as singlet, doublet, triplet, … were rapidly used to characterize the individual proton lines in



the NMR spectra and to assist the identification of the molecule under study. Realizing that these same J-couplings could be used to transfer magnetization from one spin to another, Jeener proposed in 1971 to use an indirect acquisition scheme to reconstruct a 2D map (Jeener and Alewaeters, 2016), whereby the off-diagonal peaks connect different protons through their

J-coupling. The homonuclear COSY experiment was born (Aue et al., 1976), and many two- and higher dimensional homonuclear pulse sequences would adopt a similar principle.

One problem with proton ($^1$H) NMR is the limited range of chemical shift values (typically 10 ppm) with line widths in the order of magnitude of 1 Hz, leading to a crowded spectrum especially when complex mixtures are studied. The carbon ($^{13}$C) chemical shift range is significantly larger (~ 200 ppm), making it an attractive alternative to characterize a (mixture of)

molecule(s). However, its low natural abundance (~1.1%), inherently poorer sensitivity with its fourfold lower gyromagnetic ratio and long relaxation times increases the required acquisition times by a factor of $10^6$ or higher when with a similar S/N as for the proton spectrum is desired. The introduction of a two-dimensional (2D) Heteronuclear Single Quantum Correlation (HSQC) spectrum by Bodenhausen and Ruben in 1980 has literally been a game changer in the NMR field (Bodenhausen and Ruben, 1980). In this pioneering publication, cited 3000 times as today, "the detection of NMR spectra of less sensitive nuclei

coupled to protons was shown to be significantly improved by a two-dimensional Fourier transform technique involving a double transfer of polarization." Building on the ideas of 2D NMR (Jeener and Alewaeters, 2016; Aue et al., 1976) and of magnetization transfer between a heteronucleus and proton by their scalar coupling (Maudsley and Ernst, 1977; Morris and Freeman, 1979), polarization transfer from the proton to the insensitive heteronucleus in the excitation step and back for the detection led to a tremendous gain in sensitivity. The HSQC experiment was born, and although initially described for a proton

directly linked to its amide nitrogen, it was readily applied to other nuclei such as $^{199}$Hg (Roberts et al., 1980), $^{113}$Cd (Live et al., 1985), $^{31}$P (Bolton and Bodenhausen, 1982) and $^{13}$C (Bendall et al., 1983). The HSQC experiment applied to the latter $^1$H-$^{13}$C pair has not only solved sensitivity problems associated with the direct $^{13}$C 1D spectrum but also greatly increased the information content of both individual $^1$H and $^{13}$C spectra as it connects a carbon signal directly to its proton binding partner. The success story of the HSQC spectrum was born and formed the basis for a myriad of 2- to nD spectra used from protein

NMR to the analysis of metabolomics samples.

Indeed, beyond synthetic chemistry, NMR has found its place in the realm of metabolomics (Emwas et al., 2019; Wishart, 2019), and is together with mass spectrometry the method of choice for characterizing complex mixtures such as biofluids. If throughput is required, the 1D $^1$H spectrum remains the general workhorse, whose assignment can be based on database

searches, on the spiking of relevant standards within the sample, or on analysis of the corresponding 2D spectra acquired on one or more samples. The $^1$H-$^{13}$C HSQC spectrum proposed by Bodenhausen and Ruben again is a crucial element in the latter approach, as it spreads out the proton signals according to the $^{13}$C chemical shift of its attached carbons, and allows connection of both in the same time. The use of 2D experiments in the field of metabolomics, despite inherent longer experimental times, has grown extensively, in part due to the introduction of ultrafast acquisition (Tal and Frydman, 2010), ASAP (Acceleration

by Sharing Adjacent Polarization) techniques (Schätzlein et al., 2018) and non-uniformly sampling (NUS) acquisition scheme

**MAGNETIC RESONANCE**
Discussions

(Guennec et al., 2014; Schlippenbach et al., 2018; Zhang et al., 2020). Based on these developments, the use of 2D experiments for larger number of samples has become within reach and will probably place the $^1$H-$^{13}$C HSQC spectrum as a cornerstone of metabolomics studies by NMR.

In the initial HSQC paper, it was recognized that obtaining a single peak for the $^{15}$N-$^1$H pair requires decoupling both during the indirect detection (obtained by a single proton $\pi$ pulse) and the direct proton detection (through a broadband decoupling scheme). However, for technical reasons related to the large spread of the $^{13}$C resonances, decoupling during the direct acquisition period can lead to sample heating and/or probe arcing. One typically limits acquisition times to 100ms, and especially with current high S/N cryogenically cooled probes, constructors limit the power delivery during this decoupling period (Bahadoor et al., 2021). As a result, the resolution in the $^1$H dimension is limited and the homonuclear coupling pattern of the protons cannot be observed, despite the fact that they carry valuable information.

Here, we explore the capacity to virtually decouple a HSQC spectrum where the physical decoupling during the acquisition time is omitted, thereby removing all physical limits on the acquisition time. We reconstruct the initial decoupled spectrum by recording two uncoupled HSQC spectra, one with the direct $^1$H-$^{13}$C doublet in-phase and one with the doublet in anti-phase. Comparable sequences aimed at measuring the $^1$J coupling constant have been published before, especially in the framework of determining the residual dipolar couplings for samples partially oriented in the liquid phase(Andersson et al., 1998). However, here we use both spectra only to distinguish the up- and downfield component of the doublet, thereby providing the necessary information for shifting one component with its complex homonuclear coupling pattern to the central position. As a result, we obtain a HSQC spectrum with a high resolution in the proton dimension. Traces from it can be immediately superimposed on the 1D $^1$H spectrum or on a high-resolution J-resolved (J-res) spectrum, thereby helping the assignment of the latter. We first demonstrate the approach on the spectrum of an isolated oligosaccharide, and show its use on a zebra mussel (*Dreissena polymorpha)* hydrophilic extract that we recently studied in the framework of an eco-toxicological study (Prud'homme et al., 2020).

## 2. Methods

All experiments were performed on an Avance III 800 MHz spectrometer (Bruker BioSpin GmbH, Karlsruhe, Germany), equipped with a 1.7 mm triple-resonance HCP or a 5mm quadruple resonance QCI-P (H/P-C/N/D) cryoprobe and were recorded at 298K. Data were processed with TopSpin 4.0.8 (Bruker BioSpin GmbH, Karlsruhe, Germany). All proton spectra were referenced to the methyl proton of 3-trimethylsilylpropionic acid-d4 (TSP-$d_4$). $^{13}$C chemical shifts were determined by indirect referencing(Markley et al., 1998) .





Pulse sequences were initially tested on a sample of 1 mg of dextran oligosaccharide dissolved in 40 µL of $D_2O$ plus 2 µL of TSP-$d_4$ on the 1.7 mm cryoprobe. CLIP/CLAP experiments were acquired with 16384 ($^1$H) * 256 ($^{13}$C) time points, a spectral width of 12.0172 ppm ($^1$H) * 60 ppm ($^{13}$C) and NS = 8, DS = 16, RG = 512, d1 = 1.5 s, experimental time = 1 h 23 min 2 sec

(CLIP) and 1 h 22 min 39 sec (CLAP). Data were transformed to a matrix of 16384 ($^1$H) x 1024 ($^{13}$C) frequency points. $^1$H-$^{13}$C HSQC with decoupling was recorded with 4096 ($^1$H) * 256 ($^{13}$C) time points, a spectral width of 12.0172 ppm ($^1$H) * 60 ppm ($^{13}$C) and NS = 8, DS = 32, RG = 512, d1 = 1 s, experimental time = 43 min 2 sec. Data were transformed to a matrix of 4096 ($^1$H) x 1024 ($^{13}$C) frequency points.

Hydrophilic extracts of Zebra mussel were prepared a described in (Prud'homme et al., 2020). Briefly, 50 mg of biomass were dried prior to resuspension in 50 µL of 100 mM potassium phosphate buffer dissolved in $D_2O$ supplemented with 1mM sodium azide and 0.5 mM TSP-$d_4$. From this solution, 40 µL were then transferred in the NMR sample.  CLIP/CLAP experiments were acquired with 16384 ($^1$H) * 256 ($^{13}$C) time points, a spectral width of 13.9486 ppm ($^1$H) * 100 ppm ($^{13}$C) and NS = 64, DS = 64, RG = 512, d1 = 1.5 s, experimental time = 10 h 28 min 51 sec (CLIP) and 1 h 25 min 48 sec (CLAP). Data were

transformed to a matrix of 16384 ($^1$H) x 1024 ($^{13}$C) frequency points. The 1D $^1$H, $^1$H-$^{13}$C HSQC with decoupling and the J-resolved experiments were recorded on the 5 mm cryoprobe on a sample of 200 µl. High-resolution 1D $^1$H spectrum was acquired with 32768 time points, a spectral width of 12.0172 ppm and NS = 64, DS = 4, RG = 8, d1 = 5.0 s.  The $^1$H-$^{13}$C HSQC was recorded with 2048 ($^1$H) * 256 ($^{13}$C) time points, a spectral width of 13.9486 ppm ($^1$H) * 100 ppm ($^{13}$C) and NS = 64, DS = 64, RG = 912, d1 = 1 s, experimental time = 8 h 2 min 26 sec. Data were transformed to a matrix of 2048 ($^1$H) x 512 ($^{13}$C)

frequency points. The J-resolved experiment was recorded 16384 time points a spectral width of 12.0172 ppm and NS = 128, DS = 64, RG = 912, d1 = 1.32 s for an experimental time 18 h 1 min 29 sec.

## 3. Results and discussion

As pointed out in the original paper of Bodenhausen and Ruben, without physical decoupling of the direct $^1$H-$^{13}$C interaction,

the theoretical pattern of a peak in the direct dimension ($F_2$) of the HSQC spectrum is a doublet, due to the heteronuclear coupling constant ($^1J_{CH}$ for a $^1$H/$^{13}$C spectrum). Each component of the doublet can moreover display a more or less complex coupling pattern due to homo- ($J_{HH}$) or hetero-nuclear (for example proton-phosphorus coupling, $J_{HP}$) coupling interactions involving the observed proton. The active $^1J_{CH}$ coupling can be removed by heteronuclear spin decoupling during signal acquisition and renders more easily interpretable 2D maps with a single cross peak per $^1$H/$^{13}$C pair. Decoupling however has

its drawbacks, especially when the bandwidth to be decoupled increases. Although numerous approaches have been developed to decrease the delivered power (Kobzar et al., 2004; Kupče, 2020) and hence reduce potential sample heating, limiting the acquisition time remains the first option, be it at the detriment of resolution and hence loss of intrinsic coupling pattern. Ideally,



one would want both – a decoupled $^1$H spectrum with a single peak per $^1$H/$^{13}$C pair with a high resolution leaving the intricate coupling pattern of the proton intact.

Omitting the decoupling during the acquisition is the obvious solution to the heating problem, but leads to a doubling of the number of peaks in the spectrum and thereby potentially increases spectral overlap. Moreover, the peaks resonate at ±J/2 Hz from their true chemical shift, whereby J is the active coupling at the origin of the cross peak, and can hence not directly be superimposed on the 1D proton spectrum dominated by the contribution of the $^{12}$C linked protons. For isolated peaks, however, this doublet structure can be easily recognized, and shifting both components back to their central position solves the problem.

This reconstruction of the decoupled spectrum meets problems, though, when peak overlap becomes important. One issue is whether a given peak corresponds to the down- or upfield component of the doublet, necessitating to look to the left or right for its corresponding component. For this, however, solutions have since long been around, notably in the endeavour to measure (small) coupling constants. Combining absorption/dispersion non-decoupled spectra has been used to measure coupling constants down to few Hertz for well resolved signals (Kessler et al., 1985; Oschkinat and Freeman, 1984). A 2D X-
filtered TOCSY (Wollborn and Leibfritz, 1992) or its corresponding 1D version (Nuzillard and Bernassau, 1994) were proposed to improve the accuracy of the coupling constant measurement. Other experiments such as α/β-HSQC/HMQC (Andersson et al., 1998) or α/β-HSQC-TOCSY (Koźmiński, 1999) were developed based on the concept of spin state selection (Meissner et al., 1997) to selectively observe in each sub-spectrum a single component of the coupling multiplet. However, as these methods are not applicable without directly linked protons to the carbon, workarounds have been proposed using
HSQMBC (Williamson et al., 2000), HSQC (Titman et al., 1989) supplemented with CPMG train pulses (Boros and Kövér, 2011; Kövér et al., 2006) to improve the intrinsic twisted linewidth due to the evolution of homonuclear proton-proton couplings. New pulse sequences were later developed to facilitate measurement of one-bond couplings constant by eliminating sources of lineshape distortion such as the Clean In-Phase (CLIP)-HSQC (Enthart et al., 2008). Acquiring the antiphase magnetization in a separate Clean AntiPhase (CLAP)-HSQC experiment followed by addition/subtraction of both spectra
yields high-resolution α and β-state subspectra. BEBOP (Broadband Excitation By Optimised Pulses) / BIBOP (Broadband Inversion By Optimised Pulses) (Kobzar et al., 2004; Luy et al., 2005) pulses on $^{13}$C can be used to obtain uniform performance of the experiments across the large carbon spectral width. Long-range proton-carbon coupling constants have been measured with the CLIP-HSQMBC (Saurí et al., 2013), PIP-HSQMBC (Castañar et al., 2014) and CSSF-CLIP-HSQMBC (Moreno et al., 2019).

Proton multiplet patterns are often seen as a source of increased peak overlap that needs to be addressed by a myriad of pure-shift methods (Zangger and Sterk, 1997; Foroozandeh et al., 2014; Castañar and Parella, 2015). These have been combined with HSQC sequences to alleviate extraction of coupling constants (Timári et al., 2016). However, we believe that the information contained in these patterns can be crucial towards the assignment and identification of metabolites in complex



mixtures. Herein, we combine the original CLIP/CLAP pulse sequences (Enthart et al., 2008) with virtual decoupling of a $^1$H-

$^{13}$C spectrum based on automated recognition of the α- and β- states of the multiplets and subsequent back-shifting of the high-resolution lines to their true chemical shift . We thereby obtain both a single peak per $^1$H/$^{13}$C moiety and the required high resolution for identification of its J-coupling pattern.

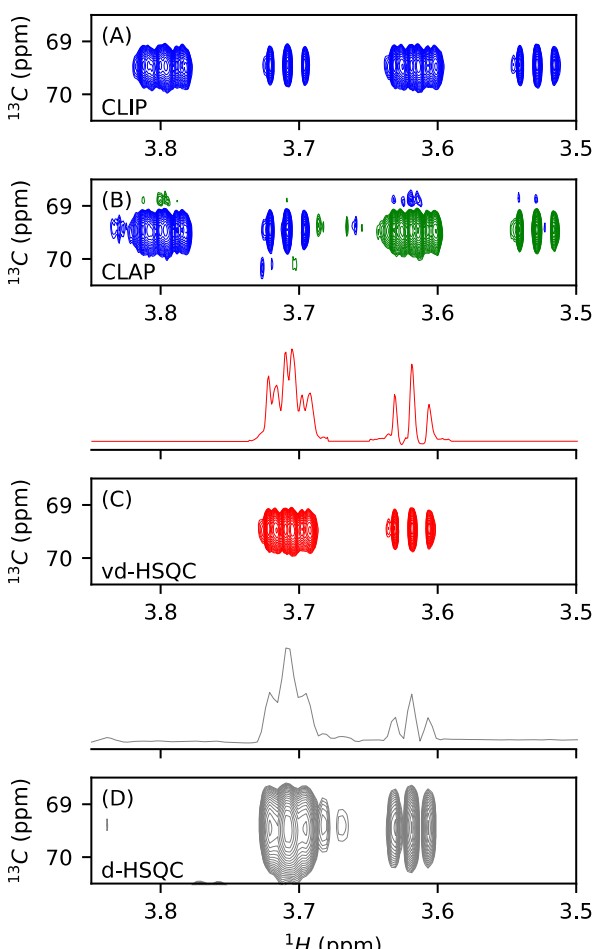

**Figure 1:** Application of virtual decoupling isolated dextran oligosaccharide. CLIP (A) and CLAP (B) shown with positive contours in blue and negative contours in green, (C) Virtually decoupled HSQC. (D) Decoupled HSQC. The traces shown above (C) and (D) are extracted at the centre of the peak of the 2D spectra.

We first tested the potential of virtual decoupling on an isolated dextran oligosaccharide. Both CLIP/CLAP experiments

acquired under identical conditions lead to two high quality spectra as shown in Figure 1 (A & B) and Figure S1. The CLIP experiment in which the antiphase proton magnetisation is refocused with a 180º $^{13}$C pulse during the final INEPT followed



by a 90º $^{13}$C pulse prior to detection to remove any dispersive component shows two purely in-phase signals separated by the heteronuclear coupling constant $^1J_{CH}$. In the CLAP experiment, these two pulses are omitted leading to the observation of the antiphase doublet. The absence of decoupling during the acquisition in the CLIP/CLAP experiment allows the observation of

each component of the doublets with high resolution (Figure S1). Taking advantage of the sign difference between the α and β components of the doublet in the CLAP experiment (Figure S2), we reduce the resolution and hence the fine structure of both components by processing the CLAP spectrum with 1024 points in the $^1$H dimension, yielding to a single peak per component. We developed a python semi-automatic procedure based on the *nmrglue* package (Helmus and Jaroniec, 2013) usable within topspin 4 that can be downloaded from https://github.com/NMRTeamTBI/VirtualDecoupling. While launching

the script from TopSpin, the user will have several options that are detailed in the Tutorial available online. The CLAP experiment is processed with 1024 time points to collapse multiplets in a single signal prior automatic peak detection. This step is performed twice: once on the positive components and once on the negative components of the spectrum and can be performed on the full spectral width or for a user defined region with a threshold that can be adapted if necessary. A clustering step based on $^{13}$C chemical shifts allows a pairwise selection of the signals. If exactly two signals of opposite sign are found

with the same $^{13}$C frequency (in points) the script will automatically bring the upfield component extracted from CLIP-HSQC spectrum with full resolution (processed with 16k in the time domain) back to the centre of the doublet therefore creating a virtually decoupled HSQC (vd-HSQC) spectrum (Figure 1.C and Figure S3). In the case of more than two signals identified at a given $^{13}$C frequency value, a user interface will allow selection and clustering of the signals.

After developing the procedure on this simple oligosaccharide sample, we turned to a zebra mussel (*Dreissena polymorpha*) extract as representative of a more complex sample. We recently annotated the spectrum in the framework of a NMR-based metabolomic ecotoxicological study and identified and assigned a large number of metabolites that can be used as reporters of environmental health (Prud'homme et al., 2020). While the initial metabolomics study of the zebra mussel (Watanabe et al., 2015) was based on 1D $^1$H and 2D $^1$H-$^{13}$C HSQC spectrum, our more recent work (Prud'homme et al., 2020) also included 2D

$^1$H-$^1$H, 2D $^1$H-$^{31}$P and J-resolved spectra and led to the assignment of ~50% of the signals observed on the 1D $^1$H spectrum. To evaluate the value of virtual decoupling on such a complex sample, we measured the CLIP/CLAP spectra of the *D. polymorpha* whole-body polar extract in order to reconstruct the vd-HSQC and combined it with a high-resolution J-res spectrum. Here, we focus on the region of the trimethylamines from 3.1 ppm to 3.3 ppm (Figure 2). The J-resolved experiment shows the presence of two singlets around 3.12 ppm (peaks A and B in Figure 2C). Because they almost overlap and moreover

differ by a factor of ten in intensity, the decoupled HSQC (Figure 2.A gray spectrum) only shows a single signal in which the two signals cannot be resolved. However, in the corresponding region of the virtually decoupled HSQC (Figure 2A) the full proton pattern containing two signals is observed, confirming that both protons are linked to a carbon with identical chemical shift. The trace extracted at 55.623 ppm through the vd-HSQC (Figure 2.B) perfectly matches the trace extracted at 0 Hz in the J-res spectrum (Figure 2.D), in agreement with a negligible isotope shift of the 13C nucleus on the proton chemical shift



(Tiainen et al., 2010). This first example illustrates the ability of the vd-HSQC to separate two signals that cannot be resolved with a conventional decoupled HSQC (d-HSQC).

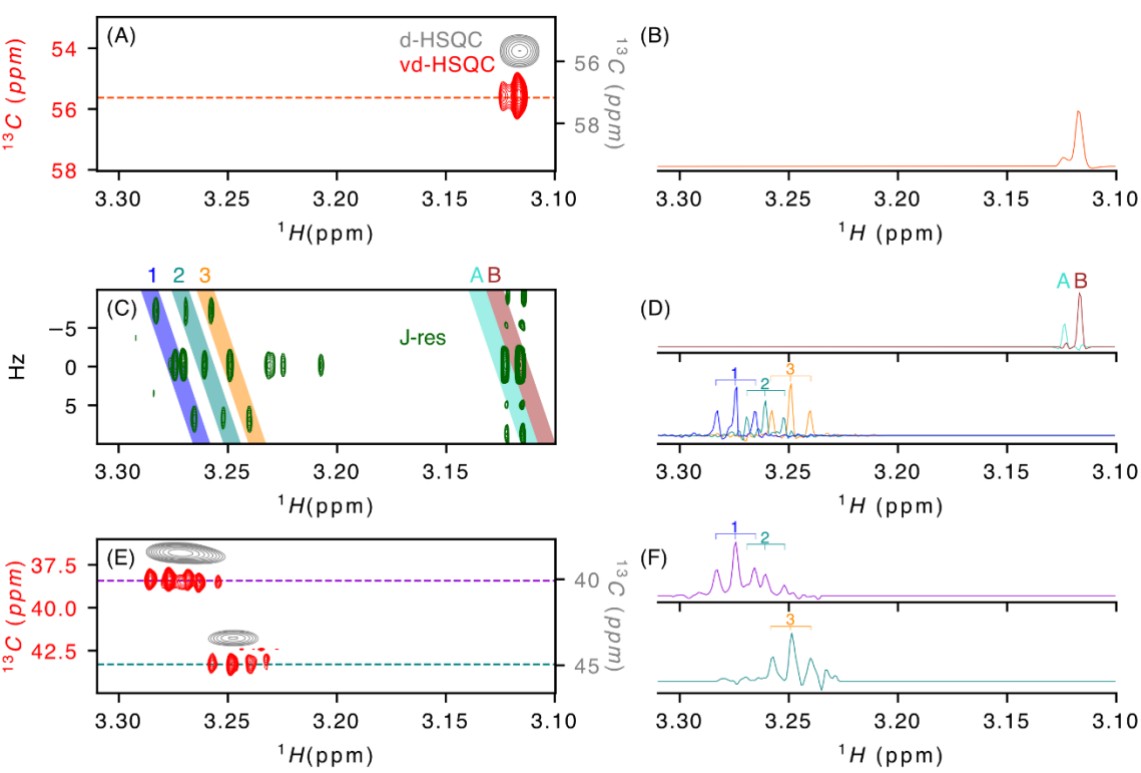

**Figure 2:** Application of virtual decoupling to Zebra mussel sample. (A) Overlay of the virtually decoupled HSQC (red) and decoupled
HSQC (gray) and (B) 1D [1]H trace extracted from the vd-HSQC at 55.623 ppm. (C) J-resolved spectrum with (D) the [1]H projections from the coloured areas on the 2D spectrum and named A/B (top) and 1/2/3 (bottom). Coloured areas were extracted using the zerter command in TopSpin **(Cox et al., 2021)**. (E) Virtually decoupled HSQC (red) and decoupled HSQC (gray) and (F) 1D [1]H trace extracted from the vd-HSQC at 38.426 ppm (top) and 43.298 ppm (bottom). 1/2/3 correspond to the three doublets of doublets identified in areas 1/2/3 of the J-res experiment.

A second example of the vd-HSQC's capacity to assist and enhance the assignment is given by the three doublets of doublets (dd) highlighted in green on the J-res spectrum (Figure 2.C). Using the *zerter* program recently developed in our group (Cox et al., 2021) first to isolate and zero the singlet at 3.27 ppm and then extract the individual pseudo-triplets, each extracted trace shows the expected 1:2:1 pattern (Lines 1, 2 and 3 in Figure 2D). These [1]H 1D traces can be used to search through the vd-HSQC spectrum to identify the corresponding [13]C resonance frequencies. Because of the high resolution of the vd-HSQC, the
scan is not only based on the chemical shift value but equally on the proton-proton coupling pattern, thereby significantly enhancing the information content. Two patterns were identified, at [13]C resonance frequencies of 38.426 ppm and 43.298 ppm (Figure 2.E). In both cases, the vd-HSQC shows the correct [1]H frequency and the correct coupling pattern. Indeed, the trace extracted from the signal at the higher [13]C frequency (Figure 2.F) shows the presence of a 1:2:1 coupling pattern which fits


that of the projected spectrum of the J-res. The pattern identified at the lower $^{13}$C frequency in the vd-HSQC shows two 1:2:1 patterns for which the $^{1}$H trace can be matched with the projection of the J-res spectrum. This combined information enhances our confidence that the three dd signals, although closely together in proton chemical shift and coupling pattern, actually represent organic moieties that differ by 5ppm in carbon chemical shift. If we compare this with a scan through the d-HSQC, assigning the broad peak at 38.426 ppm to the two dd signals seems more hazardous, especially as the singlet at 3.271 ppm could also be erroneously assigned to this proton frequency. The coupling pattern hence increases the information content and,

when combined with a high-resolution J-resolved spectrum, can lead to accurate information on individual proton resonances.

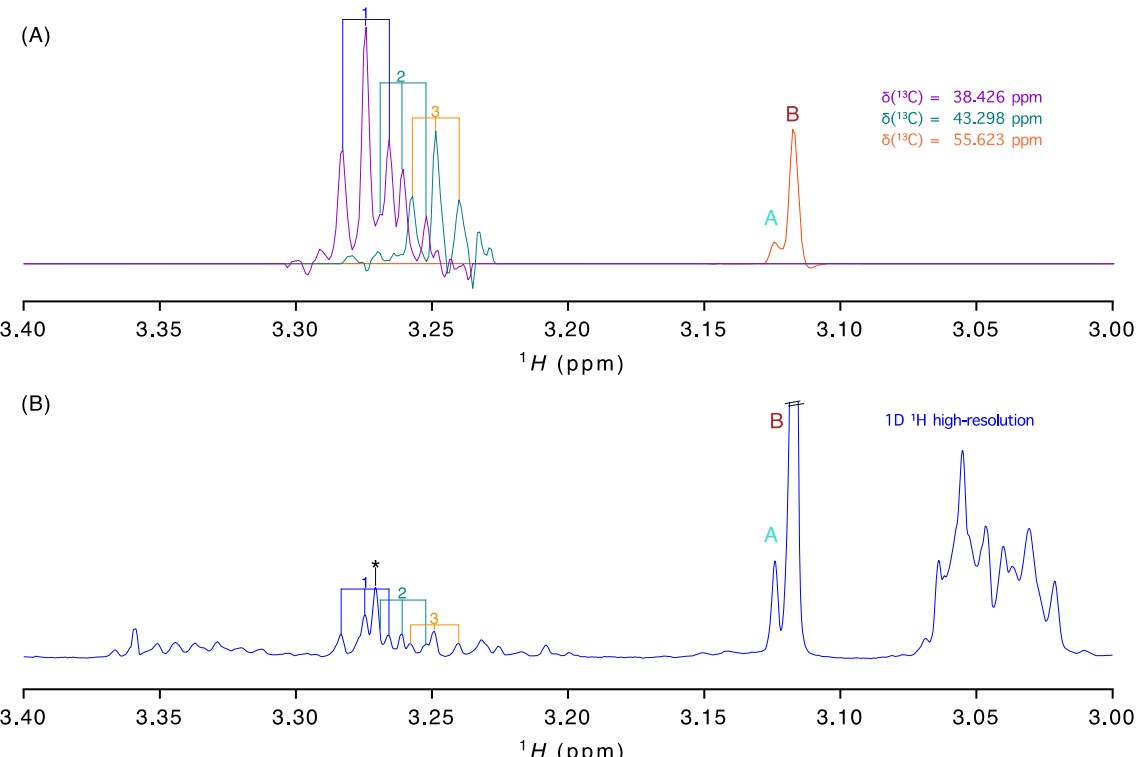

**Figure 3:** (A) $^{1}$H traces extracted from the virtually decoupling at different $^{13}$C chemical shifts (B) 1D $^{1}$H spectrum with high-resolution (acquired with 16k points). Signals are labelled as in Figure 2.


As stated before, presently most studies aiming at the analysis of a large number of samples are based on the high resolution 1D proton spectrum. To evaluate how well the information of our vd-HSQC spectrum can be transferred to the 1D spectrum, we show in Figure 3 the traces extracted from the vd-HSQC spectrum on top of the high-resolution 1D spectrum of the *D. polymorpha* extract (Figure 3). The two singlets at the $^{13}$C frequency of 55.623 ppm can immediately be identified in the 1D

spectrum, and their corresponding intensities represent the concentrations of both entities if the 1D spectrum is acquired with a sufficiently long relaxation delay. But even in the crowded region around 3.25 ppm, we can use the traces of the vd-HSQC

to assign the individual pseudo-triplets and associate them with their corresponding $^{13}$C values. Interestingly, the 1D spectrum also contains the singlet at 3.271 ppm (indicated by a * in Figure 3), but the combined information from the high resolution vd-HSQC and J-res spectra immediately singles it out as such.

## 4. Conclusion

Beyond the time constraints associated with the 2D nature of the spectrum, the use of the $^1$H-$^{13}$C HSQC in metabolomics is always associated with an NMR dilemma: broadband decoupling during acquisition greatly simplifies the spectra but due to limitations in the duty cycle leads to the loss of all information about proton multiplicities; no decoupling during acquisition leads to the observation of the complete proton multiplet pattern but doubles the number of resonances. Virtual decoupling of the spectrum is of great interest to this issue because it provides a single resonance that maintains the highest achievable resolution. Here, we explore the benefits of this approach and demonstrate that the virtually decoupled spectrum can be directly compared with the high-resolution 1D proton spectrum, thereby helping in the process of metabolite identification over many samples. The methodology presented here while being robust is rather simplistic and will benefit from the rise of deep learning and artificial intelligence applications to NMR spectra (Karunanithy et al., 2021) to provide a new entry point into metabolite databases.

**Code availability**

The code written under python for virtual decoupling and compatible within TopSpin 4 can be found on GitHub here: https://github.com/NMRTeamTBI/VirtualDecoupling.

**Author contribution**

**CC** Software, Conceptualization, Writing - review & editing; **NC, AG, J-M N, S M P** Resources, review & editing; **BL** Software, review & editing; **GL** Conceptualization, Project administration, Supervision, Writing - review & editing

**Competing interests**

The authors declare that they have no conflict of interest.

**Acknowledgements**

The authors thanks Dr. P Millard (INRAE, TBI, Toulouse) for insightful discussions. We thank Dr E Cahoreau and L. Peyriga for expert support in the MetaToul (Toulouse metabolomics & fluxomics facilities, www.metatoul.fr) NMR facility. MetaToul is part of the French National Infrastructure for Metabolomics and Fluxomics MetaboHUB-AR-11-INBS-0010



(www.metabohub.fr), and is supported by the Région Midi-Pyrénées, the ERDF, the SICOVAL and the French Minister of
Education & Research, who are all gratefully acknowledged.

270

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
