# Peer review of "Virtual decoupling to break the Simplification Versus Resolution Trade Off in NMR of Complex Metabolic Mixtures"

_Magnetic Resonance, 2021_

## Author Response (AR1)

Dr. Cyril Charlier
CNRS Research Associate
Tel : (+33) 5 31 96 89 88
charlier@insa-toulouse.fr

Toulouse, July 13th, 2021

Dear Dr. Fabien Ferrage,

We hereby wish to submit our revised manuscript entitled "Virtual decoupling to break the simplification versus resolution trade off in NMR of complex metabolic mixtures" that we would like to submit as part of the special issue "Geoffrey Festschrift" of Magnetic Resonance.

We hereby send you our revised version of the manuscript, which addresses the concerns raised by the three reviwers. In order to strengthen our study, we have included more details about the "decoupling" approach as requested by all the reviewers.

We therefore ask you to consider our revised manuscript for publication in Magnetic Resonance,

Yours sincerely,

Dr. Cyril Charlier                                          Dr. Guy Lippens

135 Avenue de Rangueil I 31077 Toulouse Cedex 4
Tél. : 05 61 55 94 01 I
www.toulouse-biotechnology-institute.fr

[Figure]

[Figure]

We thank both reviewers for their insightful and constructive comments, which have been addressed as detailed below.

**Reponse to reviewers**

**Reviewer 1:** The process of virtual decoupling is not fully clear from the manuscript. The authors refer to their software, which indeed can be downloaded from GitHub, and they provide a very brief explanation on line 180 – 189, but more details should be included in the manuscript, for example in the method section. Specifically, it is not clear if just one multiplet is 'decoupled' at a time (and how) or if the standard IPAP approach is used, with sum and differences of IP and AP followed by frequency shifts (see e.g. Figure 6 of https://doi.org/10.1016/j.pnmrs.2009.07.004).

**Response:** We have included a more detail discussion of the decoupling procedure in the the paper. Importantly, the method is at this moment semi-automatic : an interactive window allows the user to define the part of the spectrum that has to be decoupled. For the spectrum of the oligosaccharide, we immediately selected the full window, while only the multiplets of interest of the mussels sample were individually virtually decoupled. We do not use the sum and the difference of in-phase and antiphase spectra as presented by Tugarinov et *al.*. We rather use a simple approach based on the CLAP spectrum, in which we obtain the exact position of the upfield and downfield components of each doublet (positivie and negative component) detected in the desired window. Pairing of the peaks is done on the sole basis of identical $^{13}$C chemical shift. Therefore, a site specific value of J($^{13}$C-$^1$H) is obtained for each spin system which allows to overcome the wide distribution of such couplings in metabolites. The paragraph was adapted as follows :

We developed a python semi-automatic procedure for the virtual decoupling that is based on the *nmrglue* package (Helmus and Jaroniec, 2013) and is usable within topspin4. It can be downloaded from https://github.com/NMRTeamTBI/VirtualDecoupling. While launching the script from TopSpin, the user will have several options that are detailed in the Tutorial available online. First, taking advantage of the sign difference between the $\alpha$ and $\beta$ components of the doublet in the CLAP experiment (Figure S2), we reduce the resolution and hence the fine structure of both components by processing the CLAP spectrum with only 1024 points in the $^1$H dimension, thereby intentionally destroying the high resolution J coupling pattern and obtaining just a single peak per component. Automatic peak detection is then performed twice on the user selected zone, once on the positive components and once on the negative components of the spectrum, and can be performed on the full spectral width or only for a user defined region with a threshold that can be adapted if necessary. A clustering step based on $^{13}$C chemical shifts obtains a pairwise selection of the signals. The central position of this pair is used to define the J/2 value for this particular 1H/13C pair, thereby individualizing the back-shift and avoiding the impossible use of a common value for all peaks. If exactly two signals of opposite sign are found with the same $^{13}$C frequency (in digital points), the script will automatically bring the upfield component extracted from CLIP-HSQC spectrum with full resolution (processed with 16k in the time domain) back to the centre of the doublet, and thereby create a

[Figure]

135 Avenue de Rangueil I 31077 Toulouse Cedex 4
Tél. : 05 61 55 94 01 I
**www.toulouse-biotechnology-institute.fr**

[Figure]

[Figure]

virtually decoupled HSQC (vd-HSQC) spectrum (Figure 1.C and Figure S3). In the case of more than two signals identified at the same $^{13}$C frequency value, a user interface will pop up and allow manual selection and clustering of the signals.

**Reviewer 1:** The one-bond J(13C-1H) may vary from site to site. The authors should briefly describe how varying 1JCH could affect the presented virtual decoupling approach.

**Response:** This point has been answered above.

**Reviewer 2:** HSQC is not usually quantitative but I wonder if, whith this approach, the peak volume information could also be included in the metabolomics anlayses of complex mixtures. Perhaps it could be discussed in the text.

**Response:** The quantitative nature of the HSQC spectrum is not only limited by the requirement of relatively short relaxation times (as is the case for the 1D spectra), but also by the range of j-coupling values (that roughly correlate with the $^{13}$C chemical shift values). Unfortunately, this latter aspect is not addressed by the CLIP- or CLAP-HSQC, so we do not recover the quantitative nature unless when using a reference spectrum of the compound. This was added in the text.

However, for quantitative studies, the 1D proton spectrum with a long relaxation delay will remain the reference, as both limits on the relaxation time and variable J-coupling values make the HSQC spectrum inherently non-quantitative. Assignment of the 1D spectrum on the basis of the HSQC spectrum and extracting quantitative information from the former is however one way to proceed.

**Reviewer 2:** Selective cross-polarization could also be used to alleviate multiplet complexity. Obviously this is at the expense of dimensionality but if the main goal of the experiment is the chemical shift assignment, it could be eventually useful. Moreover, Prof. Bodenhausen has been very active in developing such methodology.

**Response:** We agree with the reviewer that selective $^{13}$C pulses can give 1D spectra that are equivalent to extracted lines from the HSQC spectrum. However, the problem of decoupling remains the same, and virtual decoupling then should be used to reduce this coupled 1D spectrum to a central pattern that can be compared to the 1D proton spectrum.

**Reviewer 3:** Unfortunately the approach is not obvious from the description in the manuscript - a software is mentioned where spectra are first processed at low resolution for peak picking, followed by some undescribed approach for the high resolution spectrum. I assume that both

135 Avenue de Rangueil ‖ 31077 Toulouse Cedex 4
Tél. : 05 61 55 94 01 ‖
**www.toulouse-biotechnology-institute.fr**

[Figure]

[Figure]

[Figure]

[Figure]

spectra are shifted and the size of the shift is determined by initial peak picking. As couplings in metabolites vary massively there is no common shift that works for all. The paper would be better if this process was clearly described. What are the limitations? Does this work for overlapping peaks (of species with different or similar coupling constants)? Can this also be used for $^{13}$C-labelled metabolites where adjacent $^{13}$C-atoms can lead to long range $^2$JCH couplings? What is the sensitivity of the experiment - this is crucial for often dilute metabolomics samples.

**Response:** We thank reviewer 3 for his comments which were addressed in the response of reviewer 1 & 2.

We agree that the application of our method in the case of fluxomics for $^{13}$C-labeled metabolites was not in our goal while developing it. Indeed even partial $^{13}$C labeling will greatly increase the complexity of the mutiplet in the $^1$H dimension through the long-range $^1$H-$^{13}$C couplings and can display moreover a $^{13}$C-$^{13}$C coupling in the carbon dimension.

The sensitivity of our procedure is typically half that of the HSQC experiment, as only the in-phase or anti-phase components are observed. However, the sensivity of the non decoupled HSQC spectra does benefit from the acquisition of 16k points in the $^1$H dimension.

[Figure]

135 Avenue de Rangueil ǀ 31077 Toulouse Cedex 4
Tél. : 05 61 55 94 01 ǀ
**www.toulouse-biotechnology-institute.fr**